# Eighty Years of Targeting Androgen Receptor Activity in Prostate Cancer: The Fight Goes on

**DOI:** 10.3390/cancers13030509

**Published:** 2021-01-29

**Authors:** Eva Estébanez-Perpiñá, Charlotte L. Bevan, Iain J. McEwan

**Affiliations:** 1Biochemistry and Molecular Biomedicine, Institute of Biomedicine of the University of Barcelona (IBUB), Edifici Helix, Carrer Baldiri Reixac 15-21, 08028 Barcelona, Spain; evaestebanez@ub.edu; 2Imperial Centre for Translational & Experimental Medicine (ICTEM), Imperial College London, London W12 0NN, UK; charlotte.bevan@imperial.ac.uk; 3Institute of Medical Sciences, University of Aberdeen, Foresterhill, Aberdeen AB25 2ZD, Scotland, UK

**Keywords:** androgen receptor, prostate cancer, antiandrogens, androgen ablation therapy

## Abstract

**Simple Summary:**

Prostate cancer is the second most common cancer in men world-wide, with nearly 1.3 million new cases each year, and over the next twenty years the incidence and death rate are predicted to nearly double. For decades, this lethal disease has been more or less successfully treated using hormonal therapy, which has the ultimate aim of inhibiting androgen signalling. However, prostate tumours can evade such hormonal therapies in a number of different ways and therapy resistant disease, so-called castration-resistant prostate cancer (CRPC) is the major clinical problem. Somewhat counterintuitively, the androgen receptor remains a key therapy target in CRPC. Here, we explain why this is the case and summarise both new hormone therapy strategies and the recent advances in knowledge of androgen receptor structure and function that underpin them.

**Abstract:**

Prostate cancer (PCa) is the most common cancer in men in the West, other than skin cancer, accounting for over a quarter of cancer diagnoses in US men. In a seminal paper from 1941, Huggins and Hodges demonstrated that prostate tumours and metastatic disease were sensitive to the presence or absence of androgenic hormones. The first hormonal therapy for PCa was thus castration. In the subsequent eighty years, targeting the androgen signalling axis, where possible using drugs rather than surgery, has been a mainstay in the treatment of advanced and metastatic disease. Androgens signal via the androgen receptor, a ligand-activated transcription factor, which is the direct target of many such drugs. In this review we discuss the role of the androgen receptor in PCa and how the combination of structural information and functional screenings is continuing to be used for the discovery of new drug to switch off the receptor or modify its function in cancer cells.

## 1. Introduction

Prostate cancer (PCa) is the second most common cancer in men world-wide, with nearly 1.3 million new cases each year [1]. Over the last ten years this equates to nearly 12 million men being diagnosed with PCa. It is estimated that, in the USA, 1 in 41 men will die of PCa [2] (and over the next twenty years the incidence and death rate are predicted to nearly double [3]. Interestingly, while this increase is seen in most regions of the world, it is predicted to be highest in Latin America/Caribbean, Africa and Asia [3]. Other than genetic factors a number of risk factors have been proposed for the increase in PCa: diet and age, as a result of increased life expectancy, have been particularly highlighted. The real consequence behind these statistics is that increasing numbers of men will be living with this disease and there will be an increased burden on health providers and resources.

PCa is not a new disease and there are an increasing number of detailed analyses of archaeological samples that reveal evidence of metastatic PCa in the ancient world (Figure 1) [4,5,6,7]. The most compelling evidence comes from molecular and microscopy studies on the preserved remains of a Scythian king, discovered in Siberia and presumed to have died of PCa [7]. Schmidt-Schultz and co-workers reported on the presence of bone lesions consistent with invasive growth or enzymatic activity and strikingly showed increased levels of PSA (in complex with 1-antichymotrypsin) [7]. Apart from the technological achievement of this study, it is particularly noteworthy that the same changes associated with PCa in modern patients were identified in a 2700-year-old man.

The Venetian physician and anatomist, Niccolò Massa (1495 to 1569) is credited with the first true description of the prostate gland in the 16th century. However, due to the location of the prostate gland, access to surgically excise tumours was seen as particularly challenging and from the mid-18th century to the early 20th century different approaches were trialled. Due to pioneering approaches and the skill of early surgeons such as Eugene Fuller (New York), Peter J. Freyer (UK), Hugh H. Young (Baltimore) and H. Kuchler (Germany) radical proctectomy has been a standard of care for over 100 years (reviewed in [8]). Continuing refinements to improve patient recovery and preserve quality of life include the development of laparoscopic and robotic surgery, as well as (controversially) focussed therapy in an attempt to avoid side-effects associated with surgery [9]. Of course, surgery is only a curative option in PCa confined to the prostate gland. Metastatic disease must be treated systemically and here the role of the androgen receptor (AR) becomes paramount.

## 2. Androgen Receptor—A Key Driver of PCa and Drug Target

The differentiation and development of the prostate is dependent upon the androgen dihydrotestosterone (DHT), which can be derived directly from testosterone in prostate cells or by an alternative pathway in the embryo [10,11,12]. In a seminal paper in 1941, Charles Huggins and Clarence V. Hodges demonstrated that prostate tumours and metastatic disease were also sensitive to the presence or absence of androgenic hormones (Figure 1) [13]. Thus, reduction of circulating testosterone, by removal of the testis or treatment with a synthetic oestrogen, stilbesterol, resulted in reduction in acid phosphatase, an early biomarker for PCa [13]. The significance and influence of this study for the treatment of advanced PCa cannot be overstated.

The “classical” model for androgen action, via the AR, is illustrated in Figure 2. The AR protein resides in the cytoplasm bound to chaperone complexes, but upon hormone binding to the AR with high affinity, the receptor–chaperone complex rearranges, an intramolecular interaction occurs between the amino-terminal and carboxy-terminal ends [14,15] and importin-α [16] is recruited to translocate the AR into the nucleus (Figure 2). In the nucleus, receptor dimers bind to sequence-specific androgen response elements (AREs) in the promoter and enhancer region of target genes, such as prostate-specific antigen (PSA) and transmembrane protease serine 2 (TMPRSS2). Once bound to the chromatin, AR recruits numerous coregulatory proteins to modulate transcription, leading to cell growth and survival responses [17,18,19]. At the genome level, the AR recruits members of the basal transcription machinery, for instance TATA-box-binding protein (TBP) and transcription factor IIF (TFIIF), and also other pivotal coregulators such as different members of the p160 family of coactivators and cAMP-response element-binding protein (CREB)-binding protein (CBP) (see [18,20,21,22]). However, this “simple” monomer–dimer equilibrium transition model is being challenged by ongoing research, and important questions remain regarding the possible additional role of chaperone proteins [23,24,25,26], the cellular location of dimerization [15,27,28] and the influence of response element architecture on receptor activity. Furthermore, recent studies also show that besides monomer and dimer forms, AR exists in tetramers and higher multimer oligomers even in the absence of hormone in the nucleus [28,29,30].

## 3. Androgen Receptor Structure and Function

The AR (NR3C4, nuclear receptor subfamily 3, group C, gene 4) is a member of the steroid hormone group of nuclear receptors along with the oestrogen receptors (ERα and β isoforms, NR3A1 and NR2A2, respectively), glucocorticoid receptor (GR, NR3C1), progesterone receptor (PR, NR3C3) and mineralocorticoid receptor (MR, NR3C2). The *AR* gene is present on the long arm on the X chromosome (Xq11-12) and contains eight exons interrupted by introns of varying lengths (0.7–2.6 kb) and codes for a protein of approximately 920 amino acids. There is variation in overall length due to polymorphic amino acid repeat regions within the N-terminal region. One of these, a polyglutamine repeat, is associated with the neuronal degenerative disorder Spinal and Bulbar Muscular Atrophy (SBMA) when the glutamines number greater than 38 (normal range is 9–34 in Caucasian populations) [31]. A predominant mRNA transcript of 10–11 kb and smaller species, between 6 and 8.5 kb, have been described in human breast and prostate tissue and cell lines [32,33]. The AR protein is composed of several functional domains: the intrinsically disordered N-terminal domain (NTD), the DNA binding domain (DBD) and the ligand binding domain (LBD) (Figure 3). The NTD is coded by exon 1, the DBD by exons 2 and 3, while exons 4 to 8 encode both the hinge and LBD modules [34]. We herein provide an overview of detailed AR structure and activity, its actions in PCa, and how the combination of structural information and functional screenings has been used for drug discovery of AR modulators.

### 3.1. The N-Terminal Domain

The AR-NTD makes up more than 50% of the receptor protein and shows little or no sequence homology with even its closest relatives, the GR and PR [34]. Importantly, the NTD contains activation function (AF) 1 and is essential for receptor-dependent transcriptional activation. AR-AF1 is modular in structure and function with two overlapping sub-domains: TAU1 (amino acids 103 to 372) and TAU5 (amino acids 362 to 538) [22,35,36,37,38].

In contrast to the rest of the protein the AR-NTD is intrinsically disordered and demonstrates a high degree of structural plasticity (reviewed in [39,40]), with the AF1 region shown to have a “collapsed disordered” conformation [41]. Regions of α-helical structure have been mapped by secondary structure predictions, mutagenesis and circular dichroism [37,41,42,43] and more recently by high-resolution NMR spectroscopy [21,22,44,45] (Figure 3). Intrinsically disordered protein domains allow for the coupling of protein binding and folding, which facilitates selective interactions with multiple binding partners (reviewed in [40]). Intrinsic disorder of the GR NTD has been shown to mediate allosteric regulation of receptor function and coupling transcriptional activation or repression activity with the receptor DBD [46]. Intriguingly, in the recent cryo-EM structure of the AR-FL bound to DNA the NTD was observed to form an asymmetric ring fold surrounding the DBD and LBD, which created surfaces for co-regulatory protein binding [18]. This visualisation of the AR-NTD suggests increased structural stability in the context of the DNA bound dimer complex and/or the antibody binding used to locate the domain in the structure.

In the case of the AR the presence of the NTD has also been shown to modulate DNA binding affinity for different response elements [47]. There is also considerable evidence showing that maximal activity of the AR requires an intramolecular interaction between the NTD and the ligand-binding domain, termed the N/C interaction, that occurs when ligands bind [14,48].

### 3.2. The DNA Binding Domain

The DNA-binding domain (DBD) is the most well-conserved region between different steroid receptors and is defined by nine conserved cysteine residues. The AR-DBD has a helical-globular structure (Figure 3) in which three sub-regions can be distinguished: two zinc fingers and a more loosely structured carboxy terminal extension (CTE) [49,50]. The DBD fold is stabilized by the coordination of two zinc ions by eight of the conserved cysteines [49]. The first zinc finger is important for recognition and binding of DNA response elements: three amino acid residues form the “P-box” (Gly^578^, Ser^579^Val^582^) as part of the DNA recognition helix. Five amino acids in the second zinc cluster form the “D-box” (Ala^597^-Ser-Arg-Asn-Asp^601^), involved in receptor dimerization. The P-box residues are conserved in the GR, PR and MR leading to shared response elements (often termed hormone response elements or HREs) and potentially overlapping gene signatures. However, the AR-DBD dimerisation and the CTE are thought to contribute to specific binding to selective DNA sequences, termed androgen response elements (AREs) ([49,50] and references therein). Furthermore, a relaxed stringency in the DNA sequence has been observed to be important for chromatin binding of the AR to selective response elements [51]. These different types of AR binding sites have been shown to be important for mediating normal physiology [51,52]. They may also play a role in resistance to antiandrogen therapy through GR activity [53] and have been suggested to underpin a differential response of PCa cells to a chemotherapy agent [54].

In a mouse model, termed “specificity affecting AR knock-in” (SPARKI) it was found that loss of AR binding to such selective AREs resulted in infertility (in male animals) due to impaired sperm maturation in the epididymis [52]. More recently, Robins and co-workers reported that low concentrations of doxorubicin preferentially inhibited AR target genes associated with HREs, while genes driven by selective AREs were upregulated under these conditions [54]. In ChIP-seq studies the increased binding of the AR, in the presence of low concentrations of doxorubicin, appeared to involve a redistribution of the receptor and cooperativity with other transcription factors, notably NKX3.1, HOXB13 and the pioneer factor FOXA1. However, strikingly, no identifiable HRE or ARE element was found at these sites [54]. Taken together these studies emphasize the importance of both DNA response element sequence and synergy with other DNA-binding proteins for both tissue selective gene expression and changes in the AR gene signature seen in PCa progression.

### 3.3. The Ligand Binding Domain

The onset, development and progression of PCa depends on androgenic hormone activation of the receptor and from the early work of Huggins and Hodges the AR is recognized as a crucial drug target in the fight against this cancer. First, PCa is treated by depriving tumours of androgens such as DHT and testosterone or blocking their actions by impeding their direct binding to the ligand binding pocket (LBP) on the LBD (Figure 3). The compounds impeding or blocking the binding of androgenic hormones are called anti-androgens (Figure 4). However, the effect of this type of antagonist treatment is transient, as universally patients relapse within a few years after developing a castration-resistant form of the disease, which is usually due to increased levels of AR expression or point-mutations that cause the AR to be resistant to anti-androgens, see below.

The determination of the atomic three-dimensional crystal structures provided a framework for understanding AR function and revealed detailed molecular determinants for the recognition and binding of cognate natural ligands, allowing rational drug design for the treatment of PCa. First of all, the pioneer structures revealed the folding of the LBD of AR and exhibited its overall conserved typical/canonical nuclear receptor fold. In particular, AR LBD structure contains nine α-helices, two 3_10_ helices, and four short β-strands assembled in two anti-parallel β-sheets (Figure 3). The helices are arranged in an alpha-helical sandwich; in this particular receptor the helices H4, H5, H10-11 are disposed contiguously, and H12 is folded against the body of the domain, exhibiting a completely formed coactivator binding groove also known as coactivator binding pocket or AF-2. AR does not contain a helix 2, but a long loop linking H1 with H3. Furthermore, AR and the other oxosteroid receptors feature an amino acid stretch right after the last helix H12, which is called the F-domain, a structural element that hampers the LBD domain from dimerizing using the canonical LDB dimer architecture [30]. It is also worth noting that the AR uses a novel dimerization interface involving residues in helix 5 [27].

These early models also allowed the characterization of the cocooned ligand binding pocket (LBP) and detailed atomic information on how the hormones or analogous molecules were nested in the pocket. This initial information allowed more predictive models of designed compound binding and fuelled intense drug discovery programs to develop more selective competitive AR modulators (SARMs). Since these first structures of the LBD [55], structure-based drug discoveries in academic and pharmaceutical settings have continuously been reported. Many studies have focused on providing new insights into the mechanisms of AR-targeted compounds and action in PCa but most importantly how to design more potent and selective anti-androgens with fewer side effects that may bypass resistance.

The LBP of AR exhibits numerous residues (Figure 5), which form important contacts with the natural hormone or the ligand metribolone (R1881). A total of twenty residues interact with the agonist in the LBP. Most of the residues involved are hydrophobic and are responsible for the interaction with the steroid scaffold of the agonists. The remaining residues are polar and engage in hydrogen bonds with the polar atoms of the ligand; conserved water molecules have also been described trapped in the internal cavity [56].

Several point-mutations in the LBP (Figure 5) have been described to be associated with PCa or androgen insensitivity syndromes and the crystal structures of agonist-bound AR LBD provided a structural basis to explain their impact on the structure–function relationship underlying the receptor functionality under (patho)physiological conditions. Most importantly, the difference in some of the residues that form the LBP is what confers ligand specificity. The structures provided information on how mutations linked to PCa clustered near the position 17 beta hydroxyl group of the ligand, while mutations associated with androgen insensitivity syndromes clustered around other parts of the bound ligand. There are also some residues that are found mutated in both diseases, and mutations identified are summarised in the AR Mutations Database (http://androgendb.mcgill.ca) [57].

Mutations in the LBP have been shown to underlie a switch in activity of some therapeutic antagonists to have agonist activity. These point mutations encode residue substitutions that increase the agonist activity of the receptor because it no longer recognizes clinical anti-androgens as antagonists, and the receptor continues being activated. Several mutations inside the LBP (e.g., T878A, W742L, F877L) have been found in advanced tumours and observed to result in the acquirement of agonist activity of anti-androgens. Most of these mutations expand the LBP cavity by concrete structural rearrangements in the surrounding helices conforming the walls of the pocket. So AR LBP, as has been shown for other related receptors, exhibits certain structural plasticity and flexibility in its LBP that allows the protein to rearrange—induced by the action of designed ligands and/or point mutations that are mostly treatment-induced [58].

## 4. Targeting the Hormone-Binding Function of the Androgen Receptor

Inhibitors of AR action can be broadly categorised as drugs that target the steroid biosynthetic pathway through effects on the adrenals, gonads, hypothalamus or pituitary gland, including leuprolide, finasteride, dutasteride and abiraterone or drugs that act on the receptor protein directly, for example bicalutamide and enzalutamide [59] (Figure 2 and Figure 4).

Anti-androgens have been developed that differ in their chemical structure and exhibit different efficacy and safety profiles and in the case of flutamide, nilutamide and bicalutamide were primarily developed to be used in combination with chemical or surgical castration to provide combined androgen blockade. In subsequent clinical trials with enzalutamide, significant survival benefits were demonstrated in patients with metastatic, castration resistant PCa (mCRPC). More recently developed antiandrogens such as apalutamide and darolutamide have proven to be effective at overcoming resistance to the antiandrogens bicalutamide and enzalutamide [60,61].

The first anti-androgen to be used clinically was the steroid cyproterone acetate (CPA; Figure 1 and Figure 4), a competitive AR inhibitor, with additional inhibitory effects on the synthesis of androgens and spermatogenesis [62,63]. Steroidal AR inhibitors often display mixed agonist–antagonist activities and cross reactivity with other steroid receptors, leading to an array of side effects. Side-effects of CPA are mediated through PR and GR binding and include negative feedback on the hypothalamic–pituitary–adrenal (HPA)-axis, leading to a reduction in adrenocorticotropic hormone (ACTH) and cortisol [63]. Other steroidal antiandrogens include trimethyltrienolone (RU2956, anandron), developed as a derivative of the anabolic–androgenic steroid metribolone (R1881) [64]. However, the development of the first non-steroidal antiandrogen flutamide (Figure 1) prevented progression of this compound to the clinic.

Flutamide (Figure 1 and Figure 4) is a competitive inhibitor of hormone-binding and was the first non-steroidal AR antagonist to be approved for clinical use [65,66,67,68,69,70]. Anti-androgenic activity was shown to be due to metabolism to hydroxyflutamide [66]. However, due to hepatotoxicity, clinical preference for PCa is now bicalutamide or newer second-generation antagonists (see below). Derivatives of flutamide include nilutamide, which also acts by competitively inhibiting hormone binding was approved for use in Europe in 1987. However, again, clinical use has been limited by its toxicity profile. Moreover, in the early 1970s, a compound structurally related to thalidomide, N-(3,5-dimethyl-4-isoxazolylmethylphthalimide (DIMP), was found to have weak antiandrogen activity, but was never marketed [71].

Bicalutamide (Figure 4) was developed as a more selective anti-androgen, capable of inhibiting peripheral AR with reduced effects on negative feedback loops in the hypothalamus and pituitary, possibly due to reduced blood brain barrier penetrance [72,73]. The affinity of bicalutamide for the AR was found to be four times greater than that of hydroxyflutamide, and significantly did not cause an increase in serum testosterone. Bicalutamide progressed to clinical trials under the brand name Casodex in 1989 [74] and received approval for use by the FDA (Food and Drug Administration) in 1995 (Figure 1). Bicalutamide remains a first line therapy for the treatment of metastatic prostate cancer. However, AR point mutations have been identified, in patients and cell lines, which promote resistance to bicalutamide, including a well-characterized mutation of tryptophan 742 in the LBD [75].

Enzalutamide (Figure 4) is a second-generation anti-androgen with higher affinity for the AR than bicalutamide, and capable of retaining activity in the presence of increased AR expression, often seen in castration resistant PCa (CRPC) [76]. Enzalutamide has also been reported by some as preventing nuclear translocation of the AR, resulting in impaired binding of the AR DBD to hormone response elements in androgen responsive genes (Figure 3). With FDA approval in 2012 (Figure 1), enzalutamide has entered the clinic for therapeutic use as an antiandrogen. Despite its increased efficacy compared to bicalutamide, research has still unveiled mechanisms of resistance in patients, including the F877L mutation in the AR LBD causing an antagonist-to-agonist switch [77,78]. Resistance is also indicated, and potentially conferred, by the expression of AR splice variants lacking the LBD of which the best characterised is AR-v7 [79,80]. A long, non-coding RNA *Malat1* has also been shown to drive enzalutamide mediated selection of AR-v7 expression and downregulation of this splice variant can be achieved by siRNA and the degradation enhancer ASC-J9 [81]. The F877L mutation has also been shown to confer resistance to apalutamide (formerly ARN-509, Figure 4), which, along with darolutamide, has recently been approved for non-metastatic CRPC (Figure 1) [60,61]. It is worth noting, that darolutamide remains functional as an antagonist in the presence of this receptor point mutation, as well as a number of other mutations associated with reduced efficacy of bicalutamide and/or enzalutamide (e.g., H875Y, M896V) [82].

Non-steroidal anti-androgens compete with testosterone and DHT for AR binding, thereby inhibiting receptor-dependent gene regulation and prostate tumour growth. However, antiandrogens also act systemically throughout the body and can therefore block negative feedback of testosterone action at the hypothalamus and pituitary. This can result in a moderate increase in the levels of plasma testosterone and oestrogens, which in PCa may eventually lead to resistance. Thus, anti-androgen withdrawal is recognized to be beneficial for a sub-set of patients, providing a short-term regression in disease (up to six months). The mechanism underlying this response or indeed identification of the patients likely to show a response is poorly understood, but mutations in the AR may play a part.

## 5. New Approaches to AR Inhibition in CRPC and Associated Clinical Trials

### New Approaches to Existing Antiandrogens

Anti-androgens have now been improving outcomes in advanced PCa for decades, indeed they are moving into their third generation. Many in the field feel there is still more to be done with existing, widely-used, well-characterised anti-androgens and that moving them earlier in the pathway, or/and combining them with other drugs, will improve patients’ outcomes and quality of life further [83]. The largest trial addressing this is the STAMPEDE (Systemic Therapy in Advancing or Metastatic Prostate Cancer: Evaluation of Drug Efficacy) trial (NCT00268476), which has been running since 2005 and encompasses over 10,000 patients in the UK and Switzerland. This ongoing multi-arm, multi-stage trial recruits men who are hormone-naïve and about to start androgen deprivation therapy (ADT) for localised or metastatic PCa, allocating to them to one of the recruiting treatment arms (10 treatment arms have been trialled to date) for standard-of-care plus experimental treatment or the control standard-of-care arm (ADT plus radiotherapy for non-metastatic disease) arm. Pertinent to this review, STAMPEDE first demonstrated (with almost 1000 men per group) that addition of abiraterone to ADT improves overall survival, prolongs efficacy of ADT, and delays disease progression [84]. Notably side-effects were increased by addition of abiraterone, but these were largely controlled by addition of prednisone. Another large phase III trial, LATITUDE, reported similar benefits and adverse effects [84,85]. The anticipated outcome is that, if license is granted, men may be offered a combination of ADT and abiraterone earlier, rather than waiting until ADT alone loses efficacy.

Last year two further multinational trials, ENZAMET (NCT02446405) and ARCHES (NCT02677896), reported that addition of the antiandrogen enzalutamide to standard-of-care improved overall survival and radiographic progression-free survival, although with some toxicity [84,85,86,87], The FDA has now approved enzalutamide in the metastatic hormone-sensitive PCa setting, i.e., prior to onset of CRPC. Excitingly, a STAMPEDE trial arm combining both enzalutamide and abiraterone with standard-of-care has closed and reporting is eagerly anticipated.

As discussed below, a number of new therapies are designed to target functional domains of the AR other than the LBD. That said, the AR LBD remains a valuable therapeutic target, both because there is compelling evidence that use of a second anti-androgen or SARM after resistance develops to a first has clinical benefit, and because the more selective ligands may reduce side-effects by acting in a more tissue-specific manner. Hence new SARMs are continually being developed and assessed for their ability to extend life and, importantly, improve quality of life in CRPC. Apalutamide has recently been approved for use in non-metastatic CRPC based on results of the SPARTAN trial (NCT01946204), again combining the new anti-androgen with standard ADT. Median metastasis-free survival increased from 16.2 months in the control arm to 40.5 months in the Apalutamide arm, again with slightly increased incidence of adverse effects [88]. Darolutamide is also approved by the FDA for non-metastatic CRPC based on favourable results from the ARAMIS trial (NCT02200614) which demonstrated an increase in metastatic-free survival from 18.4 months in the control arm to 40.4 months in the darolutamide arms; risk of death was also reduced [89].

## 6. Selective Androgen Receptor Degraders (SARDs)

Extending the use of drugs that target the LBD, recent years have seen the development of a new class of inhibitors, which not only compete for hormone binding, but have the capacity to induce AR protein degradation (Selective AR Degraders, or SARDs) [90,91]. Recently, Gustafsson and colleagues described two SARDs, SARD279 and SARD033, where a non-steroidal androgen, RU59063, was linked to a hydrophobic adamantyl group via a short linker [92]. The SARDs generated were able to specifically down-regulate AR protein levels, block receptor-dependent transcription and proliferation of the LNCaP prostate cell line. Importantly, these compounds also demonstrated efficacy against the F877L receptor mutation, responsible for enzalutamide resistance. Other proposed approaches to enhance receptor turnover include use of “inhibitors of apoptosis proteins” (IAPs), which promote proteasome-dependent degradation through target protein ubiquitination [91], conjugating metallo-based cytotoxic agents such as cisplatin [93], and “Proteolysis-Targeting Chimera” (PROTAC) technology [94]. PROTACs work, similarly to IAPs, by linking the protein of interest to a recognition sequence for ubiquitin-protein ligase, and subsequent proteasomal degradation. One such compound, targeting the AR LBD via an AR ligand, is ARV-110. This PROTAC is orally bioavailable, effective in preclinical models of enzalutamide -naïve and -resistant xenograft models and is currently in first-in-man phase I clinical trials (NCT03888612) [94,95,96]. Initial results demonstrate promising clinical activity in patients with CRPC, most importantly in patients that have also previously been treated with antiandrogen drugs abiraterone, enzalutamide and bicalutamide, who showed reduced PSA levels [95,97].

Galeterone, a potent steroidal multi-target agent, has been shown to reduce AR levels by increasing degradation, reducing levels of both full-length AR and AR-v7 [98,99,100]. Galeterone has been shown to inhibit CYP17A1 as well as promoting receptor degradation and impairing target gene expression demonstrating inhibition of the AR axis at multiple stages [98,99,100]. Galeterone is also active against the LBD mutations T878A and F868L [98]. Phase I and II clinical trials have been undertaken with galeterone, showing that the drug is well tolerated and has measurable effects on PSA levels [101]. Interestingly, although galaterone is marketed primarily as an antiandrogen, it is an analogue of abiraterone, marketed as a steroid synthesis inhibitor via its action on CYP17A1. Both drugs in fact have dual activity—in vitro studies showed that abiraterone can also inhibit AR activity [102]. The anti-androgenic properties of abiraterone are thought to be, at least in part, the result of its metabolism to Δ4-abiraterone, which inhibits multiple steroidogenic enzymes (CYP17A1, 3βHSD, SRD5A) and the AR protein [103].

In recent developments, Narayanan and co-workers have described a number of small molecules, UT-69, UT-155 and UT-34, which inhibited AR transcriptional activity and led to ubiquitin-mediated turnover of the receptor protein [104,105]. While UT-69 and -155 bound to the LBD, all three SARDs were also found to interact with the AR-NTD and resulted in degradation of AR-FL and AR-Vs. UT-34 was effective against enzalutamide resistant tumours in preclinical models and demonstrated excellent pharmokinetic and safety profiles [105]. AZD3514 [106] is another SARD shown to be effective in preclinical studies at inhibiting receptor signalling and reducing tumour burden in xenograft models by targeting the AR for degradation. However, in two clinical trials AZD3514 was not well tolerated and significant side effects, including nausea and vomiting, were reported [107]. It will be interesting to see which of the SARDs is first to progress into clinical practice for PCa.

## 7. Looking Outside the Ligand-Binding Pocket

Although new antiandrogens such as apalutamide and darolutamide continue to be developed and brought to the clinic, the emergence of tumour resistance due to point mutations and, more recently, the discovery of AR variants without the canonical druggable LBD which may contribute to resistance [79,80,108] (Figure 3) has contributed to the exploration of non-LBD centred pharmacological strategies.

### 7.1. The Amino-Terminal Domain

The structural plasticity of the AR-NTD, which underpins transactivation activity, presents several challenges in terms of identifying and developing antiandrogens targeting this domain. However, in a ground-breaking study, Sadar and co-workers identified a bisphenol A glycidyl ether derivative from a marine sponge extract that exhibited anti-androgen activity [109]. The bisphenol A derivative EPI-001 was shown to bind to and inhibit the AR via an interaction in the amino terminal domain of the receptor and selectively impair recruitment of co-regulatory proteins [109,110,111]. The binding site for EPI-001 was mapped by high resolution NMR spectroscopy to the helical region, amino acids 354 to 448, containing the sequence motif WHTLF [22]. In a separate study Dehm and co-workers showed that, at concentrations of 50 μM, EPI-001 inhibited the PPARγ (NR1C3) receptor suggesting both AR-dependent and -independent mechanisms of action [112]. Importantly, EPI derivatives were capable of inhibiting the AR and reducing tumour burdens in animal models and entered a small-scale phase I/II clinical trial as EPI-506 (ralaniten) in 2016 (NCT02606123) [113]. In more recent studies ralaniten and a “next generation analogue” increased the sensitivity of PCa cells in vitro to ionizing radiation [114]. Although the clinical outcomes were promising the trial was stopped in 2018 as the required dosing was deemed a burden. Development is continuing of second generation of inhibitors based on EPI-506.

Interestingly, the Sadar group has described other marine compounds reported to mediate antiandrogen activity, naphetenone B and sinkotamides A-E, which also target the AR-NTD [113]. Sintokamides A-E are bioactive chlorinated peptides isolated from the sponge *Dysidea* sp., and inhibited proliferation of LNCaP cells and hormone dependent AR transactivation as well as forskolin induced transactivation by the AR-NTD [115,116]. Niphatenone B is a glycerol ether which inhibited receptor-dependent transcriptional activation and androgen driven proliferation of LNCaP, but not cells lacking the AR. Niphatenone B blocks AR N/C interaction and expression of AR regulated genes. However, it also covalently binds both the AR and GR AF1 regions [117,118]. The reported roles of the glucocorticoid receptor in PCa are conflicting, acting as a tumour suppressor in some circumstances, and promoting PCa progression in others. However, in CRPC cell lines lacking AR expression the GR is capable of driving AR responsive gene expression and cell proliferation, and this suggests that co-targeting of both the AR and GR might be beneficial in some instances [53]. These small molecules remain the subject of preclinical research.

In addition to the described studies screening marine sponge extracts, on-going research into targeting the AR-NTD includes high throughput screening of compound libraries (Monghan et al. In preparation), alongside the design of polypeptide antagonists based on the AR-NTD [119,120] and mechanistic insights from selective AR down-regulators (SARDs) [104,105].

### 7.2. BF3 Surface Pocket

Ligand binding creates surfaces on the LBD such as the AF2 and BF3 pockets, which directly or indirectly are important for protein–protein interactions [14,121]. These surface pockets have also been recognised as suitable drug targets for the treatment of both PCa and SBMA and a number of small molecules that modulate receptor activity in preclinical models have been identified; none have yet progressed to clinical trials [122,123,124,125,126].

### 7.3. The DNA Binding Domain

As AR-variants lacking the LBD retain the DBD, targeting therapies to the DBD is an emerging area of interest. Surprisingly perhaps, given that its structure has been known for the longest, this is the least-targeted region of the AR, probably due to its high homology to the DBD of other steroid receptors, so cross-inhibition is a potential problem. Recently, the PROTAC approach has moved in this direction: MTX-23 is a PROTAC targeted to the DBD, which recruits Von Hippel-Linda E3 ubiquitin ligase resulting in degradation of both full-length AR and AR-v7 [127]. Only tested pre-clinically to date, it was effective in inhibiting growth in models of resistance to abiraterone, enzalutamide, darolutamide and apalutamide.

Another approach to DBD inhibition is screening for small molecule inhibitors. *Pyrvinium pamoate* (PP) was in this way found to be a non-competitive AR inhibitor. While it did also inhibit related receptors, including GR and ER, interestingly the inhibitor activity appears to be highly tissue specific and highest in prostate, potentially limiting off-target effects [128]. However, to date this and other reports of small molecule DBD inhibitors [129], that also inhibit full-length AR, AR-v7 and PCa proliferation, have not progressed to clinical trials.

## 8. Combination of Antiandrogen and Chemotherapy

Despite the mixed or inconclusive outcomes of early studies with cytotoxic drugs [59] taxanes have proved highly effective and docetaxel and cabazitaxel were approved for the treatment of mCRPC in 2004 and 2010, respectively. These cytotoxic agents work by stabilizing the microtubule cytoskeleton leading to cell-cycle arrest and apoptosis in actively dividing cells. It is interesting then that trafficking of the hormone-bound AR to the nucleus (Figure 2) appears to be mediated by binding to the motor protein dynein and the microtubule network [130]. This has led to the suggestion that taxanes act in part by directly disrupting AR signalling and that examination of receptor cellular localization could provide a prognostic readout for the effectiveness of taxane therapy [130]. In a subsequent study the microtubule binding site was mapped to the AR-DBD and hinge region [131]. Strikingly, it was found that nuclear trafficking of AR-Vs was differentially sensitive to taxane treatment. AR-v7, which lacks the hinge domain, in contrast to AR-v567, did not bind the microtubule network and was refractory to docetaxel inhibition [131]. Given taxanes represent first- and second-line chemotherapy for PCa it was significant that docetaxel inhibited prostate cancer cell xenografts expressing AR-v567, but not xenografts with AR-v7 [131]. These studies further demonstrated a role for taxane inhibition of AR signalling and importantly highlighted expression of splice variants as a potential mechanism underpinning chemotherapy resistance.

The differential trafficking of AR splice variants is intriguing, as AR-FL and AR-v7 have been found to have distinct but overlapping gene signatures (see [132]). However, the absence or presence of the hinge region did not appear to alter the function of AR-Vs and Weigel and co-workers concluded that the pioneer transcription factor FOXA1 and chromatin accessibility accounted for the differential gene signatures seen for the full-length receptor and splice variants. Notwithstanding these findings, the trafficking of the AR-FL and AR-Vs is an important area of receptor biology that requires further research.

In contrast to the above study [130], Kyprianou and co-workers identified the AR-NTD as the site of β-tubulin binding (see [133] and references therein). This led to an investigation of the potential synergy between docetaxel and the NTD-inhibitor EPI-002. Growth of the prostate cell line 22Rv1, expressing both AR-FL and AR-Vs was inhibited both in vitro and in vivo (as a xenograft) by docetaxel and further inhibited by the combined treatment with EPI-002 [132,133]. In this cell model there was little or no effect of docetaxel on nuclear localization of AR-FL and AR-Vs, in contrast to LNCaP cells where nuclear localization was blocked by the taxane and/or EPI-002. Using a different cell model, resistance to docetaxel and cabazitaxel was generated in the CRPC cell line LNCaP95, leading to increased expression of AR-v7 and the target gene *UBE2C*. However, in this model system docetaxel treatment did not reduce AR-target gene expression and although selective knock-down of AR-v7 resulted in decreased cell proliferation it failed to restore sensitivity to taxane treatment [134]. Taken together the findings of these different studies suggest the impact of taxanes on AR signalling may be dependent upon the concentrations used and/or the hormone-sensitivity of the prostate cancer cell model studied. Meanwhile further studies with men with mCRPC did not support a role for AR splice variants and taxane-resistance [134,135].

## 9. The Future of Anti-Androgen Therapy

With the projected rise in the number of men with PCa there will be a continued clinical need for improved diagnosis and therapeutic options. The importance of AR signalling in advanced and mCRPC ensures that the receptor will remain a primary drug target. Whilst academic research and the pharmaceutical industry continue to develop new LBD targeted anti-androgen therapy, the mechanisms of resistance that drive CRPC suggest that the provision of fundamentally different approaches to therapy may be beneficial.

The AR-NTD is essential for transcriptional activity and despite the structural challenges of this domain has become the target for a number of candidate inhibitors and drug-discovery approaches. Similarly, targeting surfaces of the AR distinct from the LBP (AF1, AF2, BF3, DBD) while also posing significant challenges, offer the potential for developing “selective androgen receptor modulators” and novel inhibitors to overcome drug resistance that arises in advanced and metastatic PCa and for reducing side-effects. It is also worth remembering that, although most AR-focused drug discovery programs are aimed at tackling PCa, receptor modulators may also be of value for the treatment of emerging other conditions such as androgenic alopecia, breast cancer, melanoma, medulloblastoma, androgen insensitivity syndromes or SBMA. However, despite promising results for both SARMs and SARDs in cell-based and, in some cases, pre-clinal in vivo models none have so far been translated to the clinic. To date, only ralaniten (EPI-0506) the NTD inhibitor has made any real progress towards clinical development.

## 10. Conclusions

The AR is the key driver of, and therapy target in, PCa at all stages including advanced CRPC. Therapy resistance can arise via alterations in the AR, but new drugs and combination strategies can overcome this. The development of effective new drugs relies on detailed knowledge of the mechanism of action of AR, its structure and how this changes when bound to ligands. In the near future we anticipate the licensing and/or clinical adoption of both new combinations of existing drugs and of new drugs targeting different regions of the AR protein.

## Figures and Tables

**Figure 1 cancers-13-00509-f001:**
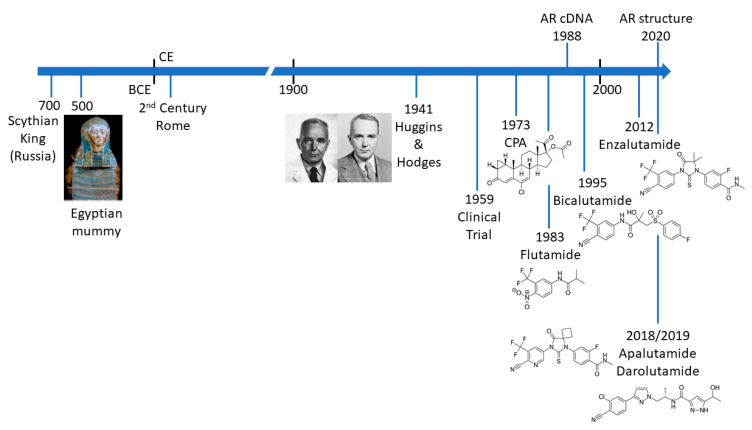
Timeline illustrating key events in the treatment of prostate cancer.

**Figure 2 cancers-13-00509-f002:**
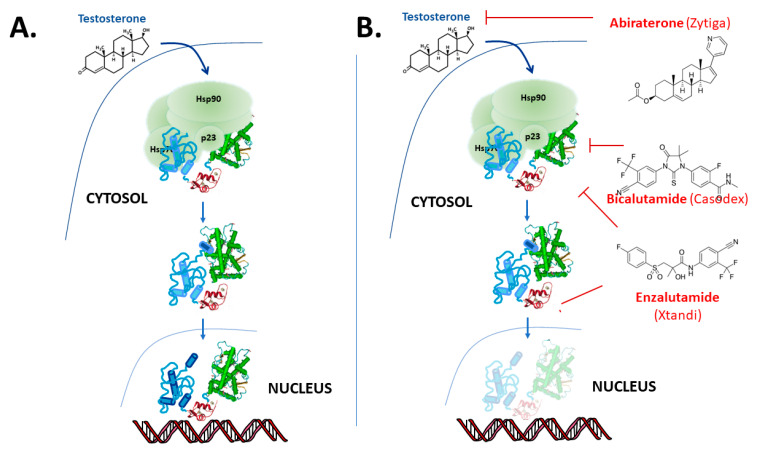
Overview of androgen receptor mechanism of action. (**A**) In the “classical” model the AR binds to testosterone and in tissues such as the prostate preferentially to the more potent metabolite 5α-dihydrotestosterone (DHT) and dissociates from molecular chaperones and translocates to the nucleus where it binds to DNA response elements and up- or down-regulates target gene expression. However, the details of a number of these steps in this model remain subject to debate and on-going research. (**B**) The action of the drugs abiraterone, bicalutamide and enzalutamide used in the treatment of PCa.

**Figure 3 cancers-13-00509-f003:**
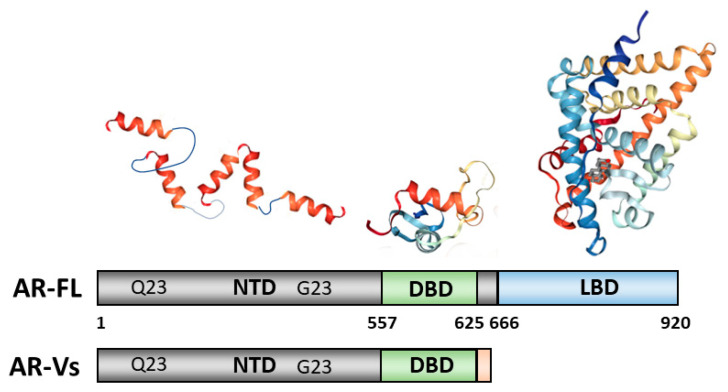
Functional and structural domains of the androgen receptor. The full-length androgen receptor (AR-FL) has a variable number of amino acids due to highly polymorphic glutamine (Q) and glycine (G) repeats in the amino-terminal domain (NTD). Splice variants (AR-Vs) lacking the ligand-binding domain (LBD), but retaining the DNA-binding domain (DBD) and NTD, are thought to emerge in PCa as a consequence of androgen ablation therapy. Available structural information for isolated ligand-binding (PDB 1I37) and DNA-binding (PBD 1R4I) domains is shown above the schematic of the AR-FL. Note—in the text the number of amino acids and point mutations is based on AR-FL of 920 amino acids.

**Figure 4 cancers-13-00509-f004:**
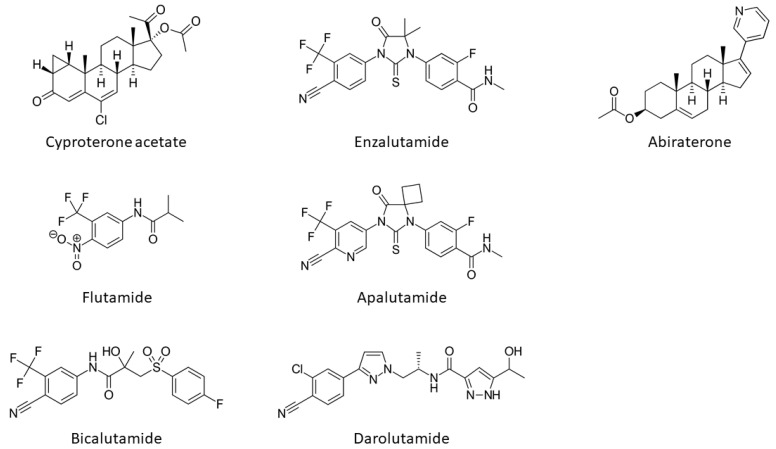
Structures of androgen receptor antagonists and the CYP17A1 inhibitor, abiraterone.

**Figure 5 cancers-13-00509-f005:**
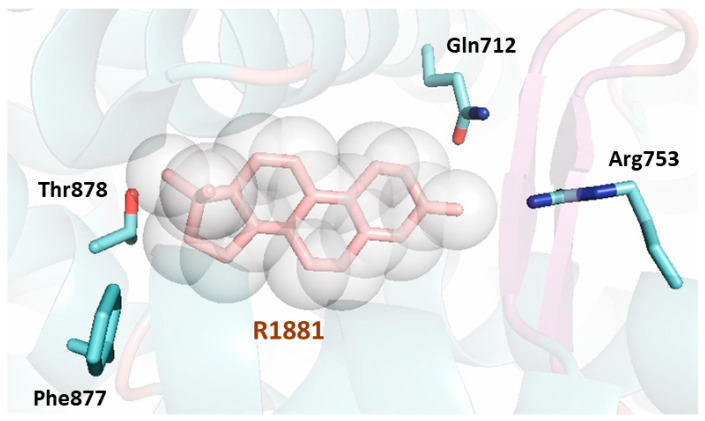
Cartoon representation of the first AR LBD crystal structure solved (PDB [Protein Data Bank] code 1E3G) [55]. The AR LBD secondary structure is shown depicting the helices, loops, and β-sheets surrounding the ligand (shown as a stick model in pink surrounded by a sphere representation). The LBP of the AR LBD is formed by twenty residues lining the bound ligand. Three residues make hydrogen bonds with the ligand (Gln712, Arg753 and Thr878) and are shown as sticks in blue showing their lateral chains. The residue Phe877, also shown as a stick representation in blue, has been found to be linked to enzalutamide resistance. All the residues lining the pocket extracted from the PISA server (https://www.ebi.ac.uk/pdbe/pisa/) are: Leu702, Leu705, Asn706, Leu708, Gly709, Gln712, Trp742, Met743, Met746, Val747, Met750, Arg753, Phe765, Met781, Met788, Leu 874, Phe877, Leu881, Met896 and Ile900.

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
