# Peer review of "Eighty Years of Targeting Androgen Receptor Activity in Prostate Cancer: The Fight Goes on"

_cancers, 2021, doi:10.3390/cancers13030509_

Round 1

Reviewer 1 Report

In the present, relatively short review article, the authors deal with the history, present status and future aspects on the androgen receptor (AR) as the therapeutic target for prostate cancer.  They appear to discuss primarily topics of their own interest that is naturally their prerogative.  In doing so, however, several important aspects of androgen action and/or antiandrogen resistance remain uncovered.  As a consequence, an uninitiated reader will not be given a state-of-the-art treatise on the current status of the field. 

In order to improve the text and render it more updated, the authors should insert multiple additions into the current typescript, as outline below.

  1. In the description of the DNA-binding domain of AR on p. 5, the authors state in a very laconic fashion that it binds to response elements often termed GREs.  However, the AR-DBD interacts with two distinct classes of response elements: one that is shared by AR and glucocorticoid, progesterone and mineralocorticoid receptors and the other one that is AR-specific (e.g., Adler et al.; 1993; Sahu et al., 2014) and that exhibits less stringent requirement for the 3’ hexamer.
  2. The above features of androgen-response elements (AREs) have importance in prostate cancer biology, in that glucocorticoid receptor (GR) signaling can activate a part of AR-dependent genetic networks and thereby bring about resistance to antiandrogen therapy (e.g., Arora et al., 2013).
  3. The consensus AREs (GREs) and AR-specific AREs seem to drive dissimilar biological responses: the former proliferation and the latter differentiation.  From prostate cancer therapy point of view, this dissimilarity appears to be important for doxorubicin response (Kregel et al., 2020) or bipolar androgen therapy.
  4. Over the last five years, a few review articles have appeared in a top-tier endocrine journal, Endocrine Reviews (Pihlajamaa et al., 2015; Zhao et al., 2019), covering mechanisms of androgen action, AR agonistic and antagonistic ligands, as well as resistance to antiandrogen treatment in a much more comprehensive fashion than in the present manuscript.  These articles need to be cited.

Minor issues that need to be addressed.

  1. The title should not start with a numeral, in that “80” should read “Eighty”.
  2. In the legend to Fig. 2, it would be appropriate to mention that AR binds preferentially to DHT in tissues such as the prostate with high 5α-reductase activity.
  3. DHT should be spelled out first time as 5α-dihydrotestosterone as opposed to dihydrotestosterone.
  4. In their discussion on AR-LBD mutants that bring about resistance to enzalutamide and/or apalutamide, the authors should mention that these mutants do not affect the antiandrogenic activity of darolutamide.
  5. Some of the citations appear to refer to meeting abstracts. Does journal policy permit this?

Author Response

Reviewer 1

We thank the reviewer for their comments and constructive suggestions. As the reviewer alludes to, and indeed appreciates, there has been an explosion of research into the origins, progression, and therapeutic treatment of prostate cancer over the last 20 years or so. Our aim was to focus on a specific aspect of targeting the androgen receptor (AR) axis from a historical perspective through to current thinking and future directions and provide an overview and primary research citations. It was not to provide an in-depth discussion of every aspect as other reviews have covered this ground. However, we have taken on-bord the reviewer’s comments when revising our manuscript.

Points 1 to 3 (lines 177 to 197). We agree our original discussion of the AR DNA-binding domain was too brief and did not give a proper sense of the complexity and dynamics of DNA binding. We have now (with additional citations) expanded section 3.2 to reflect the different types of response element recognised by the AR, the potential for the glucocorticoid receptor to substitute for AR in castrate resistant prostate cancer (CRPC) and the recent intriguing results from Kregel et al. 2020. In the latter, as the reviewer highlighted, the nature of the DNA response element impacted on the effect of the chemotherapy agent doxorubicin. This is an important observation and has potential implication for treatment options. However, it is also worth noting that detailed study of the results presented reveal differences not captured in the model presented by the authors.

Point 4. We respect the reviewer’s opinion, but feel the main task of a review is to cite the really key relevant primary sources (directing the interested reader to more information) rather than exhaustively cite other, related reviews. Further the article from Zhao et al 2019 is a broad discussion of targeting members of the nuclear receptor superfamily in cancer, which while very valuable and interesting, is in many respects outside the scope of our discussion focused on antiandrogen treatments

Minor Points

  1. Title corrected (line 2).
  2. Agreed and change has been made (line 104).
  3. Agreed and change has been made (line 104).
  4. This is an important point and we have now added this information (and citation) (lines 350-353).
  5. We checked the Instructions to Authors and this seems to be allowed. We feel that citing these abstracts is timely and gives a sense of some current areas of research and direction of travel.

Reviewer 2 Report

The review is nicely written and provided a general overview of AR function in PCa. Some parts were already covered in other reviews regarding the receptor, however, the authors included an updated view of ongoing clinical trials regarding AR inhibitors and valuable information regarding the recent cryo-EM structure of the receptor.

I only have a few minor comments for the authors:

  • In figure 3, the authors need to include PDB IDs of the used structures in their legend.
  • In paragraph 3.2. (The DNA Binding Domain), they mentioned that three amino acids form the Pbox while they gave five residues (Gly597-Ser-cys-lys-Val582), I would recommend stating the three important residues for Pbox.
  • The numbering of AR residues needs to be updated as the number changed a couple of years ago and needs to be according to the NM_000044.2 sequence. for example, the numbering of AR LBD residues has been changed and +1 needs to be added.
  • Page 7, there is a typo in figure 4 legend: “The residue Phe876. also shown as a stick representation in blue, has been found to be linked to enzalutamide resistance”. The dot needs to be removed.
  • Page 7, there is a typo in paragraph 4: “More recently developed antiandrogens such as apalutamide and darolutamide have proven to effective at overcoming resistance to the antiandrogens bicalutamide and enzalutamide”. Have proven to be
  • Paragraph 6 p10, when talking about the dual action of abiraterone, it is worth mentioning that the conversion of abiraterone to D4A may be the underlying mechanism of AR inhibition.

Author Response

Reviewer 2

We thank the reviewer for their positive comments and constructive suggestions.

  • Figure 3- the PDB ID has now been provided (line 135).
  • We have made these corrections to Section 3.2, as well as expanded this section to reflect the complexity and dynamics of AR DNA binding (line 173-174).
  • We debated this point prior to the original submission and in agreement with the reviewer we have now used the NM_00044.2 reference sequence for amino acid numbering.
  • Typos corrected.
  • This is an interesting point we had not considered, we have now discussed this in the text and included a relevant citation (Section 5 lines 442 to 445).

Round 2

Reviewer 1 Report

The authors have addressed most of my concerns in a satisfactorily fashion.

The references 49 and 50 cited in the manuscript do not provide correct information pertaining to the two types of androgen response elements. See my original review.

Reference 134 includes two separate articles.

Author Response

We thank the reviewer for highighting the error with citation 134- this has now been corrected.

Regarding the comment about references 49 and 50 -these actually do illustrate the fact that the androgen receptor recognises two different types of DNA response element. However, revisiting the reviewers earlier comments we have now included a further statement and the citation of Sahu et al (2014) in our discussion of receptor binding to DNA.